# Does Postoperative Oral and Intestinal Microbiota Correlate with the Weight-Loss Following Bariatric Surgery?—A Cohort Study

**DOI:** 10.3390/jcm9123863

**Published:** 2020-11-27

**Authors:** Tomasz Stefura, Barbara Zapała, Anastazja Stój, Tomasz Gosiewski, Oksana Skomarovska, Marta Krzysztofik, Michał Pędziwiatr, Piotr Major

**Affiliations:** 12nd Department of General Surgery, Faculty of Medicine, Jagiellonian University Medical College, 30-688 Cracow, Poland; tomasz.stefura@gmail.com (T.S.); skomarovska.o@gmail.com (O.S.); md.krzysztofik@gmail.com (M.K.); michal.pedziwiatr@uj.edu.pl (M.P.); 2Department of Clinical Biochemistry, Faculty of Medicine, Jagiellonian University Medical College, 31-501 Cracow, Poland; barbara.zapala@uj.edu.pl; 3Department of Hematology Diagnostics, The University Hospital, 30-688 Cracow, Poland; astoj@su.krakow.pl; 4Department of Microbiology, Faculty of Medicine, Division of Molecular Medical Microbiology, Jagiellonian University Medical College, 33-332 Cracow, Poland; tomasz.gosiewski@uj.edu.pl; 5Centre for Research, Training and Innovation in Surgery (CERTAIN Surgery), 30-688 Cracow, Poland

**Keywords:** obesity, bariatric surgery, sleeve gastrectomy, gastric bypass, microbiota

## Abstract

The composition of the gastrointestinal microbiota is associated with obesity. The aim of this study was to verify if, six months after bariatric surgery, patients who achieve satisfying weight-loss after sleeve gastrectomy (SG) and Roux-en-Y gastric bypass (RYGB) have a different composition of oral and intestinal microbiota in comparison with those who do not. This prospective cohort study was conducted between November 2018 and November 2020. Participants underwent either SG or RYGB and were allocated into: Group 1—participants who achieved a percentage of excess weight loss (%EWL) of ≥ 50%, and Group 2—patients with %EWL of < 50%. The %EWL was measured 6 months following surgery. At this time, oral swabs were obtained and stool samples were provided. The endpoint was the composition of the gut microbiota. Group 1 comprised 20 participants and Group 2 comprised 11 participants. Group 1 had oral microbiota more abundant in phylum Fusobacteria and intestinal microbiota more abundant in phylum Firmicutes. Group 2 had oral microbiota was more enriched in phylum Actinobacteria and intestinal microbiota was more enriched in phylum Bacteroidetes. The compositions of the microbiota of the oral cavity and large intestine 6 months after bariatric surgery are related to the weight-loss.

## 1. Introduction

Bariatric surgery is currently regarded as the most effective treatment for morbid obesity [1]. It has a potential to achieve a sustainable excess weight-loss, as well as improve obesity-related diseases, i.e., type 2 diabetes or hypertension [2]. The most commonly performed bariatric operations currently include sleeve gastrectomy (SG) and Roux-en-Y gastric bypass (RYGB) [3,4]. Various factors can influence the postoperative course and functional recovery of patients undergoing bariatric treatment, including age, type of bariatric treatment and lifestyle choices [5,6]. Multiple researchers are currently trying to identify potential patient-related criteria that would help to determine if bariatric surgery will lead to satisfactory and sustainable weight-loss [7,8,9]. Identifying such criteria, could help to decrease the percentage of unsuccessful bariatric operations and therefore, help to reduce the risk for patients who are undergoing surgical treatment for obesity and do not achieve satisfying outcomes [10]. Bacteria present in the gastro-intestinal tract can play an important role in weight-loss or in weight-gain by regulating appetite, the amount of energy obtained from food during the digestive process or human energy reserves [11,12]. Additionally, through various neural, hormonal and immunological pathways, bacterial microbiota influences the brain-gut axis and modifies the inflammatory response and metabolism [13,14]. Therefore, it could impact obesity and obesity-related comorbidities [15]. Our hypothesis was that bacteria present in the gastrointestinal tract after the surgical treatment of obesity could influence the weight-loss outcome and determine the potential success of the bariatric treatment. Moreover, identifying the bacterial profile of patients who achieved a favorable outcome after bariatric surgery could pave the way for future studies trying to develop new interventions that would modify the microbiota for optimal outcomes. The aim of this study was to verify if, six months after bariatric surgery, patients who achieve satisfying weight-loss after SG and RYGB, have a different composition of oral and intestinal microbiota in comparison with those who do not.

## 2. Materials and Methods

### 2.1. Study Design

This prospective cohort study was conducted in one academic, teaching hospital between November 2018 and November 2020. Patients were qualified for bariatric surgery using the following criteria: Body Mass Index (BMI) ≥ 35 kg/m^2^ with obesity-related comorbidities or BMI ≥ 40 kg/m^2^ [16]. Inclusion criteria were informed consent to participate in the study, age between 18 to 65 years and meeting the eligibility criteria for bariatric treatment, either for SG or RYGB. Exclusion criteria were treatment with antibiotics within 6 months after surgery (follow-up period), treatment with probiotics within 30 days prior to gathering biological material, gastrointestinal infections, inflammatory bowel disease, thyroid diseases, history of cancer (especially in the digestive tract), and immunodeficiency. There were no patients in the study group requiring antibiotic treatment in the interval between surgery and follow-up. The study was designed and described regarding all STROBE checklist points for observational studies [17].

The authors created a database of patients’ clinical details and anthropological parameters during the course of bariatric treatment and during follow-up 6 months after the initial bariatric surgery. The database included age, sex, preoperative body weight and BMI, ASA class, main comorbidities (cardiovascular diseases, arterial hypertension, respiratory disorders, diabetes mellitus type 2) and factors related to the surgical procedures, i.e., type of procedure, operative time, postoperative complications and bariatric treatment parameters 6 months after bariatric surgery: percentage of total body weight loss (%TBWL), percentage of excess weight loss (%EWL) and percentage of excess BMI loss (%EBMIL). We defined postoperative complications as adverse events occurring within six months of the procedure and presented them according to Clavien-Dindo classification [18]. Patients were divided into two groups: Group 1—patients with %EWL at least 50% (positive response) and Group 2—patients with %EWL under 50% (negative response). Favorable weight-loss cut-off point of 50% EWL at 6 months after surgery was based on previous reports concerning outcomes after bariatric procedures in the short-term [19,20].

### 2.2. Analysis of Endpoints

The first endpoint was to compare the differences between the bacterial profiles of swab samples and stool samples between patients qualified to Group 1 and Group 2, 6 months after the initial bariatric surgery (either SG or RYGB). The second endpoint was to analyze the microbiota profiles in patients who responded positively to SG or RYGB and those who responded negatively.

### 2.3. Collection of Swab and Fecal Samples

Patients were advised to fast for a least 12 h prior to gathering the biological material. During follow-up examinations 6 months after the initial bariatric surgery, oral swabs were collected using three sterile cotton swabs (4N6FLOQSwabsTM, regular tip 4473979; Thermofischer, Waltham, MA, USA). Patients were also advised to provide stool samples collected into sterile containers.

Swab samples were collected by medical doctors wearing protective clothing and sterile gloves. Feces were collected by patients who were informed previously on how to collect samples to minimize the risk of contamination. Swab samples were stored in the original swab collection container without liquid medium. Both stool and oral swab samples were frozen at −80 °C until further processing. According to our previously published protocols of the proceedings, storage at −80 °C does not affect the results [21,22]. The bacterial DNA is very stable for a very long period of time under these conditions. All procedures were performed using sterile instruments, ensuring the integrity of the biological material and without undue delay to freeze samples within a quarter of an hour after their collection. This protocol was also included in previous research [23].

### 2.4. Treatment Protocol

All patients were treated in accordance with Enhanced Recovery After Surgery (ERAS) pathway, including preoperative, intraoperative and postoperative interventions, which were described in detail in our previous articles [5,24].

### 2.5. Surgical Technique

The surgical technique of SG and RYGB was described in detail in our previous publications. To avoid plagiarism, we decided to attach references to above mentioned articles [5,25].

### 2.6. DNA Isolation, Library Preparation and Sequencing

Surfaces and equipment were decontaminated with 70% alcohol and UV radiation were used to minimize environmental contaminants. All consumables used during sample preparation and library preparation were decontaminated by UV treatment. During the bacterial DNA extraction blank controls were used. Library preparation was performed in a separate room from DNA extraction. During library preparation, no-template amplification controls were included. Filter tips and low aerosol pipettes were used. Additionally, non-redundant dual indexing was performed to prevent index swapping during sequencing. During all sample processing stages researchers were wearing appropriate clothing including clean laboratory suits, sterile gloves and face masks. DNA from fecal samples was isolated using QIAamp PowerFecal DNA Kit (QIAGEN, Hilden, Germany) and from swab samples—QIAamp BiOstic Bacteremia DNA Ki (QIAGEN). The quality and quantity of the DNA was assessed using three endpoints. We used the NanoDrop spectrophotometer to evaluate DNA purity (A260/280, A260/230), the Qubit fluorometer with the 1XdsDNA HS (high sensitivity) Assay Kit (Invitro-gen Q32854) to evaluate DNA yield (ng) and, finally, Bioanalyzer (DNA 1000). To increase the accuracy and decrease the risk of bias, three negative controls and two positive controls and ATCC standards for oral microbiome (ATCC^®^ MSA-1004™) and gut microbiome (ATCC^®^ MSA-1006™, Manassas, Virginia, United States) were included [23]. The V3 and V4 regions (using forward and reverse region-specified primers, selected from Klindworth A. et al. publication) of the 16S rRNA gene were amplified. The PCR was conducted in a 25 µL reaction volume with the following composition: 12.5 µL of 2x KAPA HiFi HotStart ReadyMix (ROCHE, Basel, Switzerland), 5 µL forward and 5 µL reverse primer and 2.5 µL (5 ng) template. We used 25 cycles of denaturation (95 °C for 30 s), annealing (55 °C for 30 s) and elongation (72 °C for 30 s). To avoid primer-dimer formation, the PCR products were semi-quantified by using Bioanalyzer DNA 1000 chips (Agilent, Santa Clara, CA, USA). The index PCR was performed in a 50 µL reaction volume using Nextera XT index kit (FC-131-1001; Illumina, IL, USA). The libraries were validated using the Qubit fluorometer with the 1XdsDNA HS (high sensitivity) Assay Kit (Invitro-gen Q32854, Carlsbad, California, United States) and Bioanalyzer DNA 1000 chips (Agilent). Purified, quantified and pooled (4 nM) amplicons were mixed with 15% of an equimolar concentration of PhiX and sequenced at 5 pM. Next Generation Sequencing (NGS) was performed with an Illumina MiSeq platform using paired-end 2 × 301 nucleotide (nt) dual-index sequencing.

### 2.7. Statistical Analysis

Statistical analysis was performed in the STATISTICA v13 package (Tulsa, OK, USA). The data was presented as: mean ± standard deviation (SD) in relation to normal distributions, or as: median (Me) and first (Q1) and third (Q3) quartile for non-normal distributions. The distribution of the studied variables was verified using the Shapiro-Wilk test. Quantitative data were analyzed with the T-student test, Mann–Whitney U test, Kruskal–Wallis, ANOVA and post-hoc testing.

We performed the taxonomic classification of 16S rRNA targeted amplicon reads using a taxonomic database. The classification was performed using the Illumina 16S Metagenomics workflow. This analysis was based on the algorithm which is a high-performance implementation of the Ribosomal Database Project (RDP) Classifier described by Wang Q. et al. [26].

Linear discriminant analysis (LDA) effect size (LEfSe) is a computational method supporting multidimensional class comparisons, with particular emphasis on metagenomic analysis. LEfSe identifies the traits (organisms, clades, operational taxonomic units, etc.) that are most likely to explain the differences observed. It is achieved by combining standard statistical significance tests with additional tests encoding biological consistency and effect significance. The effect size provided by LEfSe reveals an estimate of the magnitude of the observed difference between previously specified groups [27].

Here, we used LEfSe to identify statistically significant differences in relative abundances of oral microbiota and intestinal microbiota between samples collected 6 months after bariatric surgery. We have also conducted similar analysis considering specific bariatric operations separately (SG and RYGB).

A default cut-off value of LDA > 2.0 was used in all tests. The Kruskall-Wallis (with alpha value 0.05) test was used to analyze all features, testing whether the values in different classes were differentially distributed. The pairwise Wilcoxon (with alpha value 0.05) was used to verify whether all pairwise comparisons between subclasses within different classes significantly agreed with the class level trend. The resulting subset of vectors was used to build an LDA model from which the relative difference among classes was used to rank the features. The final output thus consisted of a list of features that were discriminative with respect to the classes, consistent with the subclass grouping within classes, and ranked according to the effect size with which they differentiate classes.

*p*-Values below 0.05 (*p* < 0.05) were considered to be statistically significant.

### 2.8. Ethical Considerations

All procedures performed in studies involving human participants were in accordance with the ethical standards of the institutional and national research committee and with the 1964 Helsinki Declaration and its later amendments or comparable ethical standards. The study was approved by the Bioethics Committee of the Jagiellonian University (1072.6120.196.2018). Informed consent was obtained from all individual participants included in the study.

## 3. Results

### 3.1. Demographic Characteristics

The study group consisted of 31 patients; 20 (64.5%) were included in Group 1 and 11 (35.5%) were included in Group 2. No participants were lost during the 6-month follow-up period of the study. A flowchart of the study is presented on Figure 1. The mean age of patients included in this study was 43.5 years ± 12.3 years. Overall, 20 (64.5%) participants were female. Median maximal weight was 132 kg (122.5–151 kg), median maximal BMI was 47.3 kg/m^2^ (43.6–52.5 kg/m^2^) and median preoperative BMI was 44.1 kg/m^2^ (40.7–48.3 kg/m^2^). We did not identify significant differences in the above-mentioned parameters between the groups. Additional demographic characteristics are presented in Table 1.

### 3.2. Perioperative Characteristics

A group of 25 (80.6%) patients underwent SG and 6 (19.4%) underwent RYGB. Overall, 26 (83.9%) patients were assigned ASA class II and 5 (16.1%) patients were assigned ASA class III. Median SG operation time was 100 min (67.5–110) and median RYGB operation time was 125 min (120–137.5). Postoperative complications occurred in 1 (3.2%) patient. The %TBWL, %EWL and %EBMIL were 30.4 ± 11.4, 55.2 ± 17.6 and 61.3 ± 20, respectively. Six months after surgery, follow-up meetings were carried out with all of the participants. The perioperative characteristics of the participants are presented in Table 2.

### 3.3. NGS Analysis

We analyzed 62 samples (31 oral swabs and 31 fecal samples) and obtained 19,611,220 reads of the 16S RNA genes. Of these, 17,779,550 (90.7%) passed positive filtering. The mean number of reads per sample was 6625 (range: 6253–52,741). Samples in which the 16S sequence length was <1250 bp, in which there were >50 wobble bases (e.g., M, R, W, S, Y, K, V, H, D, B or N) or that were only partially classified (no classification for genus or species) were filtered out of the analysis. The cut-off for the number of reads per sample was 2000. There was no need to remove any of the samples from the analysis. The characteristics of the phylogenic analysis of the oral and intestinal samples are presented in Table 3.

Comparisons of the alpha biodiversity of the oral and intestinal samples are presented in Appendix A. We used the Reny index, which depended on the parameter alpha. Alpha = 0 gives the total species number, alpha = 1 gives an index proportional to the Shannon index, and alpha = 2 gives an index that behaves like the Simpson index [24]. The beta-diversity is presented in Appendix A as the principal coordinates analysis, which shows the distances and similarities within the oral (Whittaker index = 0.023672) and intestinal (Whittaker index = 0.064449) samples.

### 3.4. Differences in the Microbiota among the Study Groups

In case of oral microbiota, we identified inter alia *Leptotrichia hongkongensis* from phylum Fusobacteria as significantly more abundant in Group 1, whereas phylum Actinobacteria was significantly more frequent in Group 2 (Figure 2) (Appendix A). Intestinal microbiota was significantly richer in *Bactoroides massiliensis* and other bacteria from phylum Firmicutes in Group 1 and in *Bacteroides fragilis* and other bacteria from phylum Bacteroidetes in Group 2 (Figure 3) (Appendix A).

### 3.5. Differences in the Microbiota in Participants Undergoing SG

Among patients undergoing SG, oral microbiota of Group 1 was significantly more abundant in phylum Fusobacteria, whereas the composition of oral microbiota among patients in Group 2 was more plentiful in phylum Firmicutes (Appendix A). The intestinal microbiota of patients undergoing SG was abundant in bacteria from phylum Firmicutes in Group 1 and in bacteria from phylum Bacteroidetes in Group 2 (Appendix A).

### 3.6. Differences in the Microbiota in Participants Who Were Undergoing RYGB

Among patients undergoing RYGB in Group 1, inter alia *Haemophilus parainfluenzae* from phylum Proteobacteria was significantly more abundant in oral cavity, whereas *Clostridium jejuni* from phylum Proteobacteria was more abundant in large intestine (Appendix A).

## 4. Discussion

The presented study is one of the first attempts at investigating the relationship between the weight-loss and microbiota present in the oral cavity and large intestine after surgical treatment of obesity. Assessment of bacterial composition was conducted using NGS 16S rRNA analysis, which is currently is one of the most cutting-edge commonly used for identifying the microbial taxa present in a given community [28]. Moreover, majority of previous studies focused only on microbiota present in fecal matter and rarely compared the outcomes between the two most frequently performed bariatric operations: SG and RYGB [29]. Investigating bacterial microbiota in oral cavity should accompany analysis of intestinal microbiota especially that in a clinical environment, an oral swab is much easier to obtain and, therefore, it is more convenient to introduce.

Identifying the optimal microbiota composition for weight-loss after bariatric surgery could form the basis for future research focused on developing an intervention aimed to modify the gastrointestinal microbiota for better weight-loss and metabolic outcomes. A meta-analysis by Dror et al. posits that microbiota manipulation for weight-loss, as well as for weight-gain, could be a successful strategy [30]. This is consistent with other published expert opinions which state that modifying microbiota is likely to provide an important therapeutic target for management of obesity and metabolic syndrome [31].

Oropharynx usually contains a high level of bacteria, such as Streptococcus mutans, Porphyromonas gingivalis, Staphylococcus and Lactobacillus [32]. The composition of microbiota in large intestine is the most abundant in bacteria from the phyla Proteobacteria, Firmicutes, Actinobacteria and Bacteroidetes [33]. The composition of microbiota in the gastro-intestinal tract is dynamic and related closely to multiple factors. It is influenced by age, lifestyle choices, geography, systemic diseases, antibiotic use, probiotics and genetic predispositions [34,35,36].

Patients achieving favorable outcomes had oral microbiota more abundant in bacteria from phylum Fusocbacteria and intestinal microbiota was more frequently inhabited by Bacteroides massiliensis and phylum Firmicutes. On the other hand, patients who did not achieve a satisfying bariatric outcome had oral microbiota rich in bacteria from phylum Actinobacteria and intestinal microbiota abundant in Bacteroidetes. We were not able to find studies with similar methodology to the presented research. Differences between groups observed in this study seem to be independent of demographic and perioperative characteristics, which were comparable between Group 1 and Group 2.

A previous study by Grembi et al. highlighted the importance of microbiota plasticity (defined as variability in the structure and composition of the microbiota measured by β-diversity metrics) in sustained weight-loss [37]. Our study, however, assessed the composition of microbiota from a static standpoint, rather than assess changing trends in biodiversity. According to Anhê et al., mechanisms associated with microbiota are involved in rapid improvements of obesity-related comorbidities and sustained weight-loss after bariatric surgery [38]. Research on animals revealed that transplant of intestinal microbiota from rodents that received bariatric surgery induced weight-loss in other animals [39]. Therefore, it highlighted the role of microbiota in achieving favorable outcomes after bariatric surgery. Also, a systematic review from 2019 mentioned a link between beneficial metabolic response to bariatric surgery and the change in microbiota composition. Authors highlight the need to conduct more human studies to further investigate this relationship [40]. Besides microbiota, other mechanisms that sustain weight-loss after bariatric surgery include hormonal changes, changes in bile acids secretion, enteroplasticity, glucose uptake, modification of taste and smell, and increased vagus nerve signaling [41].

This study is associated with several limitations. The most critical was that the study group was not particularly abundant. However, it was comparable with previous studies investigating gastro-intestinal microbiota in the context of bariatric surgery. Sample size was limited by research funding available to authors, as NGS analyses remain to be demanding in terms of financial resources. Secondly, lack of comparison with preoperative microbiota is an additional major limitation. Without this point of reference, it is difficult to assess the influence of bariatric surgery on the composition of bacteria inhabiting the gastro-intestinal tract. It is important to mention lack of randomization between SG and RYGB group, which could result in observed differences in microbiota composition. It is important to remember that microbiota, diet and weight-loss are intertwined, and finding cause-effect relationship between them remains to be hard. This study’s main focus was to determine if there is a relationship, not to identify a cause-effect relation. Additionally, we were not able to follow the remission of obesity-related comorbidities due to to organizational difficulties associated with follow-up visits in our center.

We advise researchers who will follow this subject in the future to conduct studies on a larger group of patients, preferably on a material gathered in multiple centers. They should consider investigating the correlation between remission of obesity-related comorbidities and gastro-intestinal microbiota, which would provide a novel insight into this subject. We recommend analysis of additional bariatric procedures, i.e., single anastomosis gastric bypass, laparoscopic adjustable gastric banding, etc. Concentrating on future research on interventions modifying the oral and intestinal microbiota could improve weight-loss outcomes of bariatric treatment. The authors plan to conduct a study that will investigate changes in oral and intestinal microbiota in time after bariatric surgery, including verifying the correlation between these changes and outcomes of surgical treatment for obesity and related diseases.

## 5. Conclusions

According to the results of this study, among bariatric patients, there is a relationship between the composition of oral and intestinal microbiota 6 months after surgical treatment of obesity and achieved weight-loss. This does not seem to not be influenced by characteristics of patients undergoing surgery (SG or RYGB). Patients who achieved %EWL of at least 50% after 6 months since surgery had oral microbiota more abundant in phylum Fusobacteria and intestinal microbiota more abundant in phylum Firmicutes. Among patients who achieved less favorable outcomes, oral microbiota was more enriched in phylum Actinobacteria and intestinal microbiota was more enriched in phylum Bacteroidetes.

## Figures and Tables

**Figure 1 jcm-09-03863-f001:**
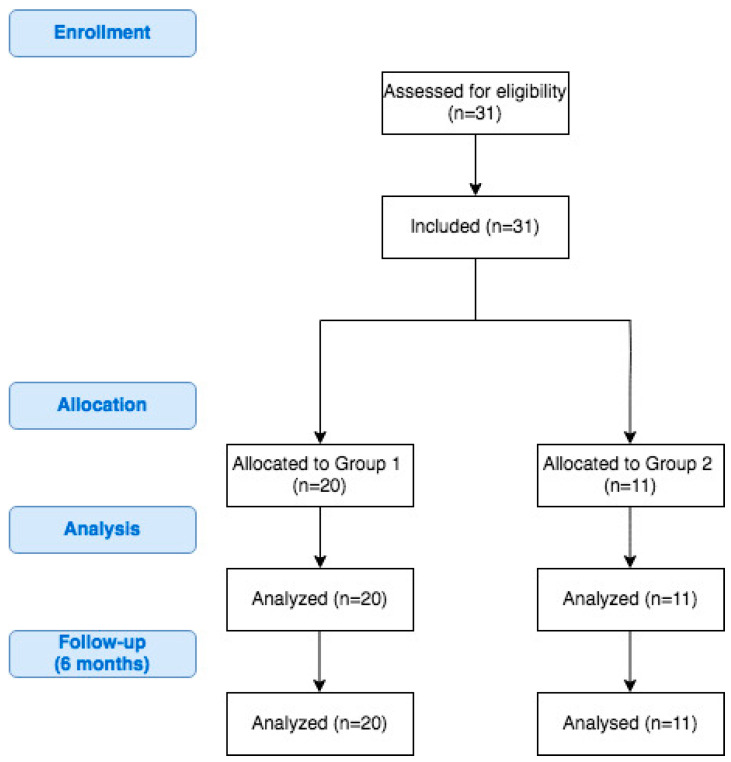
Flowchart of the study.

**Figure 2 jcm-09-03863-f002:**
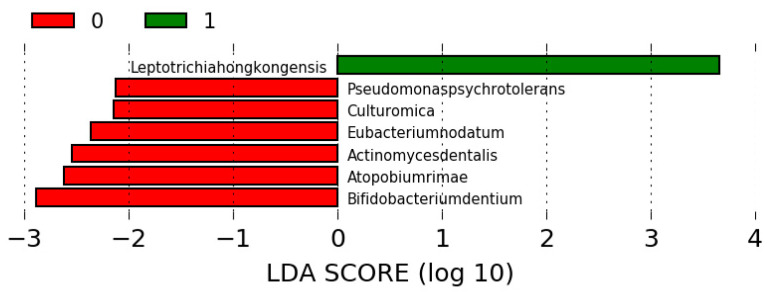
Differences in oral microbiota between Group 1 and Group 2 (1—Group 1; 0—Group 2).

**Figure 3 jcm-09-03863-f003:**
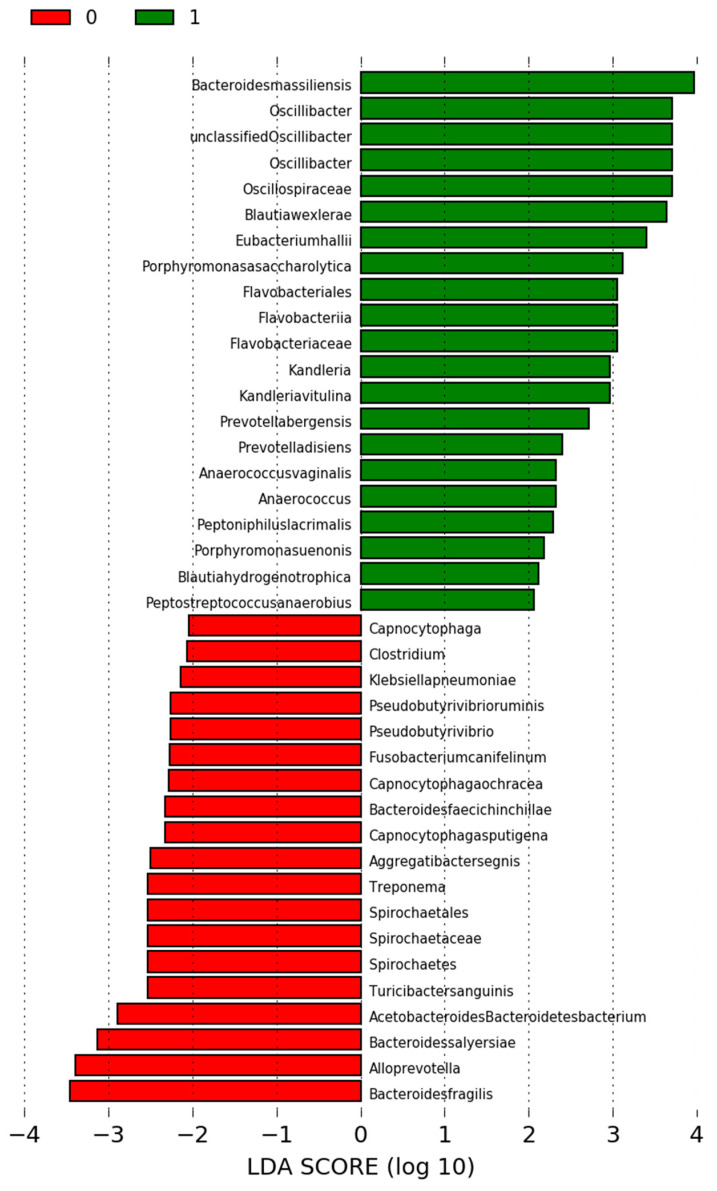
Differences in oral microbiota between Group 1 and Group 2 (1—Group 1; 0—Group 2).

**Table 1 jcm-09-03863-t001:** Demographic characteristics.

Parameter	Total	Group 1	Group 2	*p*
Total, *n* (%)	31 (100)	20 (64.5)	11 (35.5)	-
Mean age, years ± SD	43.5 ± 12.3	41.95 ± 11.01	46.46 ± 14.85	0.340
Sex (female), *n* (%)	20 (64.5)	13 (65)	7 (63.6)	0.940
Median maximal weight, kg (IQR)	132 (122.5–151)	129.5 (125.5–140.3)	132 (121.8–153.8)	0.999
Median maximal BMI, kg/m^2^ (IQR)	47.3 (43.6–52.5)	48.2 (45–50.4)	47.2 (43.1–55.1)	0.563
Median preoperative BMI, kg/m^2^ (IQR)	44.1 (40.7–48.3)	43.3 (42.7–48.7)	45.7 (40.2–47.6)	0.367
Type 2 Diabetes, *n* (%)	5 (16.1)	2 (10)	3 (27.3)	0.211
Diabetes complications, *n* (%)	2 (6.5)	1 (5)	1 (9.1)	0.657
Hyperlipidemia, *n* (%)	5 (16.1)	4 (20)	1 (9.1)	0.429
Steatohepatitis, *n* (%)	6 (19.4)	4 (20)	2 (18.2)	0.902
Hypertension, *n* (%)	22 (71)	13 (65)	9 (81.8)	0.324
Cardiovascular disorders, *n* (%)	5 (16.1)	4 (20)	1 (9.1)	0.429
Respiratory disorders, *n* (%)	4 (12.9)	4 (20)	0	0.112
Varicose veins, *n* (%)	5 (16.1)	3 (15)	2 (18.2)	0.818
Smoking, *n* (%)	5 (16.1)	3 (15)	2 (18.2)	0.818

*n*—number; SD—Standard Deviation; IQR—interquartile range; BMI—Body Mass Index.

**Table 2 jcm-09-03863-t002:** Perioperative characteristics.

Parameter	Total	Group 1	Group 2	*p*
Total, *n* (%)	31 (100)	20 (64.5)	11 (35.5)	-
Operation, *n* (%)				0.902
-SG	25 (80.6)	16 (80)	9 (81.8)	
-RYGB	6 (19.4)	4 (20)	2 (18.2)	
ASA class, *n* (%)				0.818
-II	26 (83.9)	17 (85)	9 (81.8)	
-III	5 (16.1)	3 (15)	2 (18.2)	
Median SG operative time, min. ± SD	100 (67.5–110)	102.5 (75–102.5)	90 (67.5–111.25)	0.623
Median RYGB operative time, min. ± SD	125 (120–137.5)	125 (120–160)	140 (120–132.5)	
Postoperative complications, *n* (%)	1 (3.2)	1 (5)	0	0.451
%TBWL ± SD	30.4 ± 11.4	36.3 ± 9.7	19.7 ± 3.7	<0.001
%EWL ± SD	55.2 ± 17.6	65.3 ± 12.7	36.9 ± 7	<0.001
%EBMIL ± SD	61.3 ± 20.1	71.6 ± 16.8	42.6 ± 8.6	<0.001

*n*—number; SG—laparoscopic sleeve gastrectomy, RYG—laparoscopic Roux-en-Y gastric bypass; ASA—American Society of Anesthesiologists; SD—Standard Deviation; %TBWL—percentage of total body weight loss; %EWL—percentage of excess weight loss; %EBMIL—percentage of excess BMI loss.

**Table 3 jcm-09-03863-t003:** Phylogenetic summary of results obtained.

Swab Samples
	Taxonomic Level	Abundance	Reads PF Classified to Taxonomic Level	% Reads PF Classified to Taxonomic Level
	Kingdom	2	53.20702439	99.85%
	Phylum	48	53.06839024	99.59%
	Class	85	52.95868293	99.38%
	Order	120	52.90856098	99.29%
	Family	297	52.72834146	98.5%
	Genus	1230	52.20009756	97.95%
	Species	2091	39.75509756	74.65%
Shanon-Wienner Index of diversity	3.03			
**Feacal Samples**
	Taxonomic Level	Abundance	Reads PF Classified to Taxonomic Level	% Reads PF Classified to Taxonomic Level
	Kingdom	2	55.96700000	99.85%
	Phylum	45	55.79125806	99.54%
	Class	84	55.57280645	99.16%
	Order	117	55.54338710	99.11%
	Family	285	55.09909677	98.32%
	Genus	1075	53.84638710	96.10%
	Species	1892	46.19551613	82.54%
Shanon-Wienner Index of diversity	3.54

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
