# Peer review of "Does Postoperative Oral and Intestinal Microbiota Correlate with the Weight-Loss Following Bariatric Surgery?—A Cohort Study"

_jcm, 2020, doi:10.3390/jcm9123863_

Round 1

Reviewer 1 Report

1.are you sure that the protocol to store the swab is safe without producing bis ?

2.the very low number of patients partecipating at tthis stud should be more stressed into the manuscript.

3.report a VRQ and standard accuracy of bacterial isolaton without any kind f contamination

4.what do you mean 'all procedures were carried out as soon as possible?' Time lost increase the probability to contminae the working field and specimens, and could be a signficant bias.

Author Response

Dear Reviewer #1,

Thank you for reviewing our manuscript. We have followed your suggestions and corrected the manuscript accordingly. Please find response to your comments below and in manuscript.

  1. Are you sure that the protocol to store the swab is safe without producing bias?

Thank you for your comment. The swab storage protocol was reviewed many times in our other studies [Brzychczy-Wloch M. et al., Sroka-Oleksiak A. at al.]. Storage at -80C does not affect the results. The bacterial DNA is very stable for a very long period of time under these conditions. We have added that information to the “Collection of swab and fecal samples” section of the manuscript to present a more transparent description of our Methods.

Dynamics of colonization with group B streptococci in relation to normal flora in women during subsequent trimesters of pregnancy. Brzychczy-Włoch M, Pabian W, Majewska E, Zuk MG, Kielbik J, Gosiewski T, Bulanda MG. New Microbiol. 2014 Jul;37(3):307-19

Next-Generation Sequencing as a Tool to Detect Vaginal Microbiota Disturbances during Pregnancy. Agnieszka Sroka-Oleksiak, Tomasz Gosiewski, Wojciech Pabian, Artur Gurgul, Przemysław Kapusta, Agnieszka H. Ludwig-Słomczyńska, Paweł P. Wołkow and Monika Brzychczy-Włoch Microorganisms 2020, 8(11), 1813; https://doi.org/10.3390/microorganisms8111813

2.the very low number of patients participating at this stud should be more stressed into the manuscript.

Thank you for your suggestion. We have corrected the manuscript accordingly in the limitations section to be more transparent about the limited sample size.

  1. Report a VRQ and standard accuracy of bacterial isolation without any kind of contamination.

Thank you for your comment. In frame of this study, bacteria were not isolated, but only their genetic material, using reagent kits designed specifically for extracting bacterial genetic material from swabs or fecal samples. Although it is unlikely due to selective nature of above mentioned kits, some contamination is possible. To avoid that, we used a standard protocol described by Sroka-Oleksiak A., et al. to ensure the minimization of the risk of contamination. We have corrected the” Collection of swab and fecal samples” and “DNA isolation, library preparation and sequencing” sections to describe more comprehensively measures we have taken to avoid possible contamination of the samples.

Metagenomic Analysis of Duodenal Microbiota Reveals a Potential Biomarker of Dysbiosis in the Course of Obesity and Type 2 Diabetes: A Pilot Study. Sroka-Oleksiak A, Młodzińska A, Bulanda M, Salamon D, Major P, Stanek M, Gosiewski T. J Clin Med. 2020 Jan 29;9(2):369. doi: 10.3390/jcm9020369.

4.what do you mean 'all procedures were carried out as soon as possible?' Time lost increase the probability to contminae the working field and specimens, and could be a signficant bias.

Thank you for your suggestion. We agree that the wording (‘all procedures were carried out as soon as possible’) used is incorrect. Therefore, we have changed this wording to the following: “All procedures were performed using sterile instruments, ensuring the integrity of the biological material and without undue delay to freeze samples within a quarter of an hour after their collection. This protocol was also included in previous research [Sroka-Oleksiak A., et al., 2020]”.

Metagenomic Analysis of Duodenal Microbiota Reveals a Potential Biomarker of Dysbiosis in the Course of Obesity and Type 2 Diabetes: A Pilot Study. Sroka-Oleksiak A, Młodzińska A, Bulanda M, Salamon D, Major P, Stanek M, Gosiewski T. J Clin Med. 2020 Jan 29;9(2):369. doi: 10.3390/jcm9020369.

Reviewer 2 Report

In this study, oral and colonic microbiota of two postbariatric groups – stratified by weight loss - are analyzed. 6 months postop, the composition of microbiota was related to weight loss.

The topic is very interesting and of timely relevance. I have a few questions and remarks:

Page 2, line 68: history of cancer? Or active cancer?

What about antibiotic treatment in the interval btw or and follow-up?

Stratification into groups: why 50%EWL after 6 months? As I understood from the paper, this 50% is not after 6 months.

The lack of baseline data is a problem!

As you stated yourself: composition of microbiota is very dynamic: did patients fast before or had a standardized meal?

Could composition of oral microbiota be a surrogate for type of ingested food/drinks, so that the association with worse weight loss would be explained?

Author Response

Dear Reviewer #2,

Thank you for reviewing our manuscript. We have followed your suggestions and corrected the manuscript accordingly. Please find response to your comments below and in manuscript.

Reviewer 2

Comments and Suggestions for Authors

In this study, oral and colonic microbiota of two postbariatric groups – stratified by weight loss - are analyzed. 6 months postop, the composition of microbiota was related to weight loss.

The topic is very interesting and of timely relevance. I have a few questions and remarks:

  1. Page 2, line 68: history of cancer? Or active cancer?

Thank you for your comment. A history of cancer was the excluding criterion. We have corrected the manuscript accordingly.

  1. What about antibiotic treatment in the interval btw or and follow-up?

Thank you for your comment. Although, antibiotic treatment in the interval between surgery and follow-up was not a part of the exclusion criteria in the protocol of the study, there were no patients in the study group requiring antibiotic treatment in that period. We have added that information to the study design section of our manuscript, to provide a more transparent description.

  1. Stratification into groups: why 50%EWL after 6 months? As I understood from the paper, this 50% is not after 6 months.

Thank you for your comment. We have decided to conduct a follow-up after 6 months since the bariatric procedure. Longer, for instance, 12-month follow-up period would more likely introduce bias from other factors influencing the microbiota. It would be more difficult to correlate microbiota with outcomes of the surgery, that occurred a year earlier. For example, there would be a higher risk, that patients will undergo an antibiotic treatment.

As we mention in the Study design section, presented %EWL was measured during follow-up appointment 6 months after surgery. In literature, %EWL of 50% defines successful bariatric surgery at different time points. We have chosen 6-months period to better correlate microbiota with changes in weight occurring due to surgery.

  1. The lack of baseline data is a problem!

Thank you for your comment. We have restructured our limitations section to more highlight this disadvantage of our study.

  1. As you stated yourself: composition of microbiota is very dynamic: did patients fast before or had a standardized meal? Could composition of oral microbiota be a surrogate for type of ingested food/drinks, so that the association with worse weight loss would be explained?

Thank you for your comment. As we have written in the manuscript, composition of microbiota “(…) is influenced by age, lifestyle choices, geography, systemic diseases, antibiotic use, probiotics and genetic predispositions [31–33]”. Therefore, we agree that diet could have a major impact on the composition of microbiota, as well as on weight-loss. Patients were advised to fast for 12 hours prior to gathering the biological material. Samples were collected in the morning. However, the patients did not have a standardized diet as it would be difficult to manage and control during 6 months’ period. To avoid inaccuracies, we have added information concerning fasting to the “Collection of swab and fecal samples” section. We believe, that those three factors (microbiota, diet and weight loss) are intertwined and cause-effect is very hard to determine. We have additionally added that information to the limitations section of the manuscript. We hope you will be satisfied with our explanation.